# Multimodal Deep Learning-Based Prognostication in Glioma Patients: A Systematic Review

**DOI:** 10.3390/cancers15020545

**Published:** 2023-01-16

**Authors:** Kaitlyn Alleman, Erik Knecht, Jonathan Huang, Lu Zhang, Sandi Lam, Michael DeCuypere

**Affiliations:** 1Chicago Medical School, Rosalind Franklin University of Science and Medicine, Chicago, IL 60064, USA; 2Division of Pediatric Neurosurgery, Ann and Robert H. Lurie Children’s Hospital of Chicago, Chicago, IL 60611, USA; 3Department of Neurological Surgery, Northwestern University Feinberg School of Medicine, Chicago, IL 60611, USA; 4Malnati Brain Tumor Institute of the Lurie Comprehensive Cancer Center, Northwestern University Feinberg School of Medicine, Chicago, IL 60611, USA

**Keywords:** machine learning, deep learning, multimodal, brain tumor, glioma, radiomics, genomics, prognostication

## Abstract

**Simple Summary:**

Primary malignant tumors of the brain are relatively rare, but their contribution to death due to cancer is disproportionately large. The use of multimodal data in machine learning techniques (such as deep learning) is still relatively new, but its implications for predicting brain tumor characteristics, treatment response, and patient survival are robust. In this study, we sought to review the current state of glioma prognostication using deep learning methods. A systematic review of the deep learning-based prognostication of gliomas was performed in accordance with PRISMA guidelines. All included studies focused on the prognostication of gliomas, and predicted overall survival, overall survival along with genotype characteristics, and response to immuno-therapy. Multimodal analyses were varied, with 6 studies combining MRI with clinical data; 6 studies integrating MRI with histologic, clinical, and biomarker data; 3 studies combining MRI with genomic data; and 1 study combining histologic imaging with clinical data. Overall, the use of multimodal data in deep learning assessments of gliomas leads to a more accurate prediction of overall patient survival as compared to unimodal models. As data collection and computational capacity expands, further improvements are likely from the continued integration of different data modalities into deep learning models.

**Abstract:**

Malignant brain tumors pose a substantial burden on morbidity and mortality. As clinical data collection improves, along with the capacity to analyze it, novel predictive clinical tools may improve prognosis prediction. Deep learning (DL) holds promise for integrating clinical data of various modalities. A systematic review of the DL-based prognostication of gliomas was performed using the Embase (Elsevier), PubMed MEDLINE (National library of Medicine), and Scopus (Elsevier) databases, in accordance with PRISMA guidelines. All included studies focused on the prognostication of gliomas, and predicted overall survival (13 studies, 81%), overall survival as well as genotype (2 studies, 12.5%), and response to immunotherapy (1 study, 6.2%). Multimodal analyses were varied, with 6 studies (37.5%) combining MRI with clinical data; 6 studies (37.5%) integrating MRI with histologic, clinical, and biomarker data; 3 studies (18.8%) combining MRI with genomic data; and 1 study (6.2%) combining histologic imaging with clinical data. Studies that compared multimodal models to unimodal-only models demonstrated improved predictive performance. The risk of bias was mixed, most commonly due to inconsistent methodological reporting. Overall, the use of multimodal data in DL assessments of gliomas leads to a more accurate overall survival prediction. However, due to data limitations and a lack of transparency in model and code reporting, the full extent of multimodal DL as a resource for brain tumor patients has not yet been realized.

## 1. Introduction

Brain tumors pose a substantial morbidity and mortality burden globally, affecting 330,000 people in 2016 [1]. While primary malignant tumors of the CNS are relatively rare, their contribution to cancer mortality is disproportionately large. High-grade gliomas (WHO grade 3 or 4) are the most commonly diagnosed primary malignant brain tumor, accounting for 78% of cases [2]. The prognosis for malignant gliomas remains poor, despite advancements in molecular characterization and data availability for these tumors [3]. The development of tools that can accurately predict prognosis, using multimodal data, could aid in decision-making for targeted therapies and designing treatment plans.

In recent years, remarkable deep learning (DL) advances have yielded greater capabilities than traditional machine learning methods [4]. DL is based on artificial neural networks arranged in multiple, non-linear layers, each of which perform calculations based on input from the previous layer. Multilayered calculations allow for the amplification of important aspects of input, while simultaneously suppressing irrelevant variations [4,5].

DL research has made substantial advances in various medical fields. Several studies have demonstrated the applications of DL in clinical situations, such as predicting real-time risk scores for mortality, diagnostic evaluations, and assisting in medical error reduction [6]. For instance, a recent study demonstrated a model that predicts automated breast cancer diagnosis and early-stage lymph node metastasis from whole slide images (WSI) [7]. Another study illustrated an integrated model that can predict Alzheimer’s disease progression using multimodal time series data [8]. These results have illustrated the enormous impact that DL may have on medical analysis, diagnosis, and prognosis in the coming years.

Despite the extraordinary applications of DL models, they are currently rare in clinical practice. Integration of DL models into practice faces several barriers, including a high complexity leading to limited interpretability and the potential for algorithmic biases [6]. In addition, DL model performance is dependent on the availability of training data, which may vary across clinical settings [4].

As data collection and availability continues to improve, DL may prove a useful prognostic tool for patients with gliomas [5]. Many previous DL glioma prognostication efforts have centered around a unimodal analysis of radiological imaging data [9,10]. The increasing availability of digitalized WSI, detailed phenotyping and comprehensive multi-omic datasets, such as The Cancer Genome Atlas (TCGA), provides an opportunity to integrate multimodal data into DL models. An integrated approach can provide a more comprehensive analysis than single-modal approaches and is a promising way to improve clinical performance [11]. However, as interest in multimodal methods increases, ongoing efficacy evaluation is needed to ensure patient safety. Here, we present a systematic review of multimodal DL methods for glioma prognosis and assess their effectiveness.

## 2. Methods

We systematically reviewed the literature on multimodal DL applications for brain tumor prognosis, in accordance with the Preferred Reporting Items for Systematic Reviews and Meta-analyses guidelines [12]. The Embase (Elsevier), PubMed MEDLINE (National library of Medicine), and Scopus (Elsevier) databases were queried using search terms pertaining to prognosis, deep learning, and medical imaging. This search strategy is described in full in the Appendix A.

Following duplicate record removal, title and abstract screening was performed, followed by a full-text review for final inclusion. Study inclusion was based on the following inclusion criteria: peer-reviewed original research, full length article, English language, full text available, and the use of a DL model within a workflow using multimodal data to the perform prognosis of a glioma. DL was defined in accordance with LeCun et al. [4] and included any neural network model with at least one hidden layer. Multimodal data were defined as any dataset including imaging data (e.g., MRI, WSI), in addition to one other form of clinical data, such as genomic data or clinical variables.

Screening was independently performed by two reviewers (K.A. and E.K.). Any disagreements were reconciled by discussion. Data were extracted per prespecified criteria by two reviewers (K.A. and E.K.). Due to heterogeneity in the included studies, data were synthesized qualitatively to characterize the applications and outcomes of multimodal DL methods for brain tumor prognosis. The prediction model’s risk of bias assessment tool (PROBAST) was used to assess the risk of bias for the included studies (L.Z.) [13]. PROBAST uses 20 signaling questions across 4 key domains (participants, predictors, outcome, and analysis) to determine the risk of bias of studies that develop clinical outcome prediction models. Apart from item 4.9, which evaluates the assignment of predictor weights in the final model and is not applicable to DL models due to their complexity, all signaling questions were assessed to perform the risk of bias determination.

## 3. Results

The literature search was performed on July 8, 2022 and it identified 767 unique records (Figure 1). Of these, 87 full text articles were reviewed and 16 were included in the final analysis. Table 1 summarizes the characteristics of these studies. Eight (50%) studies originated from North America, 7 (43%) from Asia and 1 (6%) from Europe. The most frequently represented countries were the United States of America (7, 43%), China (2, 12.5%), India (2, 12.5%), and Singapore (2, 12.5%).

All included texts studied adult primary glial tumors (gliomas). The most common tumor grade studied were WHO grade 4 gliomas, known as glioblastoma multiforme (GBM; 8 studies, 50%). There were 6 (37.5%) studies that included both high-grade glioma (HGG; grade 3–4) and low-grade gliomas (LGG; grade 1–2), followed by exclusively LGG (2 studies, 12.5%). The most common predicted outcomes were overall survival (13, 81%), followed by overall survival with genotype prediction (2, 12.5%) and finally the response to immunotherapy risk score (1, 6%).

The primary data sources were external databases (11, 68%), followed by institutional databases (6, 38%) and 1 (6%) study that used both. The external databases used were BraTS, The Cancer Imaging Archive (TCIA), and The Cancer Genome Atlas (TCGA). The median dataset size was 177. Diagnostic imaging was used by 14 studies in combination with one or more data types, including genomics, biomarker presence, methylation status, and clinical data. One study used histologic imaging in combination with clinical data, while the remaining study used both diagnostic imaging and histologic imaging.

In 14 studies (87.5%), the development and training of a DL model was performed, while 2 studies (12.5%) used pretrained DL models. Convolutional neural networks (CNN) were used in 12 studies (75%), 7 of which used existing DL architecture, such as U-net, VGG-16, and ResNet-18. Two studies (12.5%) used artificial neural networks (ANN) and 1 study (6%) used a fully convolutional neural network (FCNN). Training hardware was reported in 8 studies (50%) and hyperparameters were provided in 7 studies (44%).

### Integrated Multimodal Data Types

MRI and Clinical data: We identified 6 studies [14,15,16,17,18,19] that fused MRI imaging with patient clinical data to predict overall survival. The clinical data included age, gender, resection status, and tumor location. One study [13] also used treatment type, including chemoradiation, chemotherapy, targeted molecular therapy, and immunotherapy. The primary metrics used to evaluate the DL models were accuracy [14,15], AUC [16], hazard ratio [17], and cross-validation [18]. Multimodal models were compared to unimodal models in 2 of the 5 studies [16,18] and both showed improved performance over a unimodal approach. One study [19] integrated MRI and clinical data (age, gender, weight, and tumor size) to predict tumor genotype of four biomarkers (MGMT, 1q/19q, IDH, and TERT) and OS time. They then integrated the genotype features into the OS prediction model, improving OS time prediction accuracy.

MRI and Genomic data: There were 3 studies [20,21,22] that fused MRI imaging and genomic data to predict overall survival. All genomic datasets were taken from TCGA. Two studies [20,22] included the expression data of 1740 genes while 1 study [21] used a DL model to extract 20,530 genetic features from the TCGA dataset, then integrated the features with imaging data. All three of the integrated models showed a superior performance to unimodal models.

Histologic Images and Clinical Data: One study [23] utilized whole slide histologic imaging and clinical data to predict the prognosis and IDH mutational status. Clinical data included age, gender, extent of tumor resection, and tumor grade. The model achieved a C-index of 0.715 (95% CI; 0.569, 0.830) for predicting prognosis and an AUC of 0.667 (0.532, 0.784) for predicting the IDH mutation status in grade 2 gliomas.

MRI with multiple data types: There were 6 studies [24,25,26,27,28,29] that integrated MRI data with multiple different data types, such as histologic, clinical, and biomarker data.

Braman et al. included MRI imaging, WSI, genomic, biomarker, and clinical data to predict the overall survival risk score using a pretrained VGG-19 CNN [24]. The genomic data included mutational and copy number variant status, the clinical data included patient demographics and treatment type, and the biomarker data included IDH-mutant and 1p/19q-codeletion status. This model predicted overall survival with a median C-index of 0.788 ± 0.067, which considerably outperformed the best performing unimodal model with a C-index of 0.718 ± 0.064 (*p* = 0.023).

Choi et al. integrated MRI scans, biomarker data, and clinical data to calculate overall survival (OS) and the progression-free survival (PFS) prediction via an integrated time-dependent area under the curve (iAUC) [25]. The genetic data included of the IDH1 mutation and MGMT promotor methylation status, while clinical data included age, gender, treatment types, and tumor location. The performance improved when radiomic data was added to the clinical model (iAUC: OS, 0.62–0.73; PFS, 0.58–0.66) and the biomarker model (iAUC: 0.59–0.67; PFS, 0.59–0.65); however, the combined model (radiomics, biomarker and clinical) showed superior OS and PFS prognostic performance (iAUC: 0.65–0.73; PFS, 0.62–0.67).

Kazerooni et al. used MRI, biomarker status (MGMT methylation), genomic data (next-generation sequencing of 45 genes including BRAF alteration), and clinical data (age, gender, resection status) to classify patient survival time into high risk (<6 months) and low risk (>18 months) groups [26]. The multimodal model had a superior performance when compared to unimodal models.

Jeong et al. integrated radiomic data (MRI and AMT-PET), biomarker expression (IDH1 mutation and MGMT methylation status), and clinical data (age, Karnofsky performance status and resection status) to predict the 6-month PFS outcome [27]. When compared to unimodal models, the multimodal model showed a superior performance predicting the 6-month PFS (0.86 sensitivity, 0.63 specificity).

Li et al. integrated MRI scans, the presence of genomic biomarkers (immunophenoscore-associated mRNAs, MHC-related molecules, immune checkpoints, immunomodulators, and suppressor cells) and clinical data (age, gender, treatment type, and tumor grade) to calculate a risk score predictive of a patient’s response to immunotherapy [28].

Mi et al. used MRI scans, temporalis muscle area, MGMT methylation status, and clinical (age, gender) data to predict overall survival [29]. The study used 2D U-net CNN to compute the temporalis muscle cross-sectional area, then classified patients into high and low risk groups.

The risk of bias was assessed in four domains, and the overall risk of bias was low in 7 studies (44%), unclear in 3 studies (19%), and high in 6 studies (37%) (Table 2).

## 4. Discussion

More than ever, multimodal DL is being used to predict glioma characteristics and, more recently, as a tool for predicting prognosis. Overall, multimodal DL models provided more accurate survival predictions, compared to unimodal DL approaches. The most common multimodal DL combination of radiomics and genomics improved accuracy when compared to predictions based on only radiomic or genomic data. Furthermore, inputting more information, such as clinical features, immune cell markers, MGMT methylation status, and pathology data, into the DL models yielded even more accurate survival time predictions. Bimodal predictions outperformed unimodal predictions, and as more “-omics” were fused, survival predictions improved. As clinical data collection continues to expand in parallel with the computational capacity to analyze it, further improvements are likely to result from the continued integration of different data modalities in DL models.

### 4.1. Applications of Multimodal DL

The use of multimodal data in DL is still relatively new, but its implications for predicting tumor characteristics, treatment response, and survival outcomes are robust. While the use of DL for glioma segmentation from imaging is well-studied, adding other components, in conjunction with deep radiomic features, is a new development and an important area for ongoing research.

Our review suggests that fusing multiple types of data into a DL model is most useful for predicting survival outcomes. A better understanding of the tumor type and genetic features from a DL model also have the potential to aid in decision making; this is in regard to treatment options to pursue, by informing the selection of possible chemotherapy agents based on tumor-specific targets. For example, the DL model DrugCell simulates 1235 tumor cell lines’ responses to 648 drugs, as well as various drug combinations using genomic and pharmacologic data [30]. Additionally, the fusion of radiomic and clinical data, using DL, can provide an additional option for gaining detailed information regarding tumor characteristics. Overall, a better understanding of tumor characteristics in general via multimodal DL, drawing on multiple types of data from a single patient, can lead to a more optimal treatment plan and can potentially improve a variety of associated outcomes. In the case of non-small cell lung cancer, a PET/CT-based deep learning model served as an effective tool for determining EGFR mutational status and could serve to predict whether a patient will have a longer progression-free survival, in response to EGFR-TKIs or ICIs, based solely on imaging data [31].

However, there is still work to be performed to ensure that the use of multimodal data in DL makes accurate assessments of gliomas. Although there are many benefits to implementing this approach in the clinical setting, areas of improvement and development in an implementation strategy remain. There are additional data types yet to be included in DL algorithms that may yield even more accurate and detailed predictions of tumor characteristics. Moreover, further work is needed to compare multimodal and unimodal strategies head-to-head between the different DL models that have been reviewed. Certain methodologies may be better suited to the incorporation of multiple data modalities, yet there is great heterogeneity among DL strategies, limiting direct comparisons between studies. Further exploration is needed to elucidate more optimal areas of research. Growth in this area of DL has the potential to provide critical information for a more personalized treatment approach.

### 4.2. Use in the Clinical Workflow

Upon implementation in the clinical workflow, the multimodal DL approach may serve as an effective tool in segmenting and diagnosing a patient’s specific type of brain tumor. DL analysis of imaging, in conjunction with a radiologist’s interpretation, can provide a more detailed report in certain cases [32]. However, DL performance is not yet able to match to the diagnostic expertise of radiologists and should not be used as a replacement, but rather as an additional tool to improve accuracy. Jiang et al. outlines the many functions DL can serve in tumor pathology for all types of tumors, ranging from proper subtyping to prognosis prediction. However, these studies require validation, correction, and the oversight of a pathologist [33].

The use of DL to analyze deep features of scans can aid in the precise identification of the tumor type, already leading to a stronger understanding of the diagnosis before surgery and pathology results. The incorporation of a real-time automatic intensity-modulated radiotherapy DL model, which exhibited good plan quality and efficiency when used for prostate cancer, could be developed for use in brain tumor patients [34]. This has the potential to reduce the workload, improve workflow, and lead to more timely care for the patient.

### 4.3. Challenges for Implementation in Clinical Workflow

A large barrier to the clinical implementation of multimodal DL methods is the lack of data availability and reporting, which impedes clinical validation. Of note, two included studies omitted key information concerning data selection, as well as model development and evaluation, resulting in unclear determinations of the risk of bias. The potential risk of bias should always be carefully considered by researchers and clinicians when considering the published literature. The varying levels of risk of bias, among the included studies, highlights the need for the uniform adoption of reporting standards to ensure integrity and reproducibility of results [5].

Although many of the machines trained in this study utilized the BraTS database, they also included data from other institutions, including imaging, genetic information, and clinical data, not all of which have been published. The patient cohorts from institutions were typically small. Furthermore, most of the code and models used were not published, hindering the further testing of those models using different data by other researchers. Multimodal DL methods require large amounts of additional data, especially in studies involving triple and quadruple fusions. Not all of this information is included in the BraTS dataset, leading researchers to rely on data from their own institutions, which can be incomplete and lead to a low amount of training and testing data.

The BraTS dataset has played an instrumental role in the development and refinement of DL for studying gliomas. The yearly BraTS challenge calls for submissions regarding the use of novel methods (many of them DL) to segment and, more recently, predict MGMT promoter methylation status using their imaging dataset. Further additions to databases may include expanded detail, regarding factors known to feature heavily in clinical outcomes; these include the histological type of tumors, comprehensive surgical, chemotherapeutic, and radiotherapeutic data, and the molecular characteristics of tumors. More high-quality data resources such as this can lead to significant improvements in the lack of data and encourage new developments in DL to be tested in these challenges.

Even though datasets, such as BraTS, are helpful, they do not include the full range of information necessary for a multimodal approach. For example, datasets do not include protected health information and such additional data may be useful to feed into the DL algorithm. The integration of clinical data in DL has the capacity to provide accurate survival predictions up to 30 years in the future, as evidenced by the novel DL method Multiserv, which input six different types of data and was used to predict survival for 33 types of cancer [35]. Increased collaboration and the sharing of data, models and codes among researchers is necessary to overcome this limitation and enhance the performance and use of multimodal DL in the care of patients with gliomas.

### 4.4. Limitations

There are several limitations to this review. This search focused on the use of DL for the prognostication of gliomas, based on multimodal data. Thus, studies using DL for tumor segmentation, typing, and genetic predictions, based on imaging data, were not included in this review. The focus on prognostication was selected to investigate potential for direct impact on clinical decision-making. Additionally, since there is a wide and ever-evolving range of DL methods and associated terminology, the literature search may not have captured all the articles that may otherwise have been relevant to this review. Finally, the PROBAST tool, employed to determine the risk of bias, was not developed solely for the DL studies. Bias among raters may vary with respect to their backgrounds and the risk of bias assessment is subjective.

## 5. Conclusions

Overall, the use of multimodal data in DL assessments of gliomas leads to a more accurate overall survival prediction. Improvements are further compounded with the inclusion of additional types of data sources fed into the DL algorithm, such as fusing radiomic, genomic, and clinical data. The determination of a more accurate prognosis is not the only benefit of multimodal DL, as certain mutations can be identified via non-invasive means and may therefore inform treatment. The full extent of using multimodal DL as a resource for glioma patients is not realized: barriers include data limitations and a lack of transparency in regards to model and code reporting. Additionally, a strong clinical workflow needs to be established for proper implementation. Additional research into optimizing data and model combinations for more accurate survival predictions in glioma patients is clearly needed. Inputting other types of data, not reviewed in this paper, into the DL approach could also improve accuracy. Finally, multimodal DL utilization may also lend improvements to the prediction of other outcomes, such as predicted response to a specific drug or drug class. As data collection grows more comprehensive and models evolve in sophistication, multimodal DL approaches have tremendous potential to improve, via the enhanced integration of clinical knowledge.

## Figures and Tables

**Figure 1 cancers-15-00545-f001:**
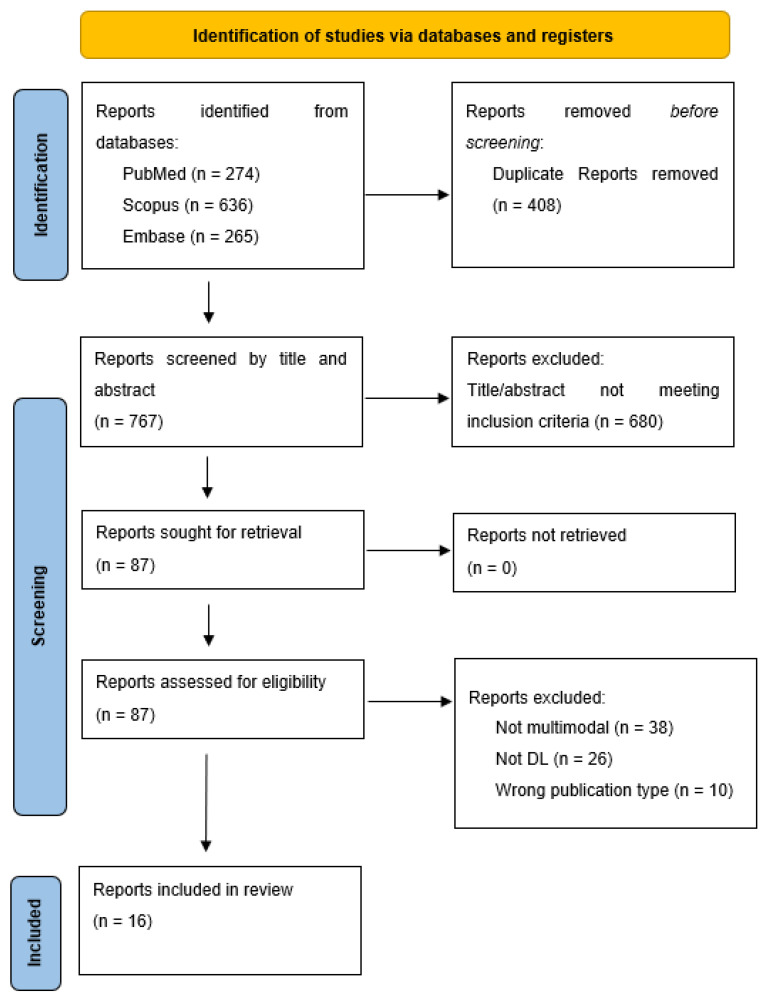
PRISMA Flowchart.

**Table 1 cancers-15-00545-t001:** Summary of studies reviewed.

Author	Year	Country	Disease	Procedure	Outcome	Predictors	Data Source	Size	Model Type	Key Findings
Alex	2018	India	Glioma (LGG, HGG)	Patients treated independently	Overall Survival Time	Radiomic, clinical data	BRaTS	241	FCNN	After using FCNN for segmentation the BraTS 2017 validation data and test data, the regressor accuracy was 52% and 47%, respectively.
Asthana	2022	India	Glioma (LGG, HGG)	Patients treated independently	Overall survival	Radiomic, clinical data	BRaTS	989	U-net	U-Net-based semantic segmentation of tumors and the pervasive learning model to calculate the weights of the regression model had accuracies of 64.2% on Brats 2018, 59.8% on Brats 2019, and 60.5% on Brats 2020 datasets.
Braman	2021	United States	Glioma (LGG, GBM)	Patients treated independently	Overall survival with predicted risk score	Radiomic, histologic, genomic, and clinical data	TCIA	176	Deep Orthogonal Fusion model, multiple-input CNN, SNN, pre-trained VGG-19	The multimodal deep orthogonal full fusion model (rad, path, genetics and clinical data) outperformed various combinations of unimodal, pairwise and triple fusion models (except for the rad, path and genetics triple fusion MMO loss).
Chaddad	2022	Canada	Glioma (LGG, GBM)	Patients treated independently	Overall survival	Radiomic, clinical data	The Cancer Imaging Archive (TCIA)	151	3D CNN	Combining DRFs (using 3D CNN), clinical features and immune cell markers as input to a random forest classifier discriminated between short and long survival outcomes.
Choi	2021	South Korea	Glioblastoma	Patients treated independently	Overall survival via iAUC	Radiomic, genetic, clinical data	Institutional	120	CNN	When CNN radiomics was combined with clinical and genetic prognostic models for overall survival and progression free survival in glioblastoma patients, the prognostic value increased.
Fathi	2022	United States	Glioblastoma	Preoperative mpMRI followed by surgical resection	Overall survival	Radiomic, genomic, MGMT methylation, clinical data	MRI scans from the hospital of the University of Pennsylvania between 2006–2018	516	VGG-16 CNN	The survival prediction performance was highest in the fusion model, combining clinical data, MGMT methylation, radiomics, and genomics, with a c-index of 0.75 and an IBS reduction of 24.8%
Han	2020	United States	Glioblastoma	Maximal surgical resection and radiation therapy (w/temozolomide or bevacizumab)	Overall survival	Radiomic, clinical data	World Health Organization IV GBM, TCGA	178	CNN (VGG-19 for deep features)	Using radiomics and CNN deep learning features extracted from GBM MRIs for a machine learning-based statistical analysis allowed for discrimination between short and long-term survivors.
Islam	2021	Singapore	Glioblastoma	Patients treated independently	Overall survival	Radiomics, genomic data	TCGA-GBM	285	FCN, cGAN, SVM, ANN	Performance almost doubled after fusing genomic features with radiomic and SVM model outperforms ANN model.
Jeong	2019	United States	Glioblastoma	Resection and subsequent chemoradiation	Progression free survival	Radiomics, clinical data	PET database at Children’s Hospital of Michigan	21	U-net	Glioma delineation by PET-based deep learning and clinical multimodal MRI data achieved the highest AUC (0.66) for survival outcome prediction.
Jiang	2021	United States	Glioma (grades 2 and 3)	Patients treated independently	IDH mutational status and overall survival	Histologic, genetic, clinical data	TCGA	296	End-to-end deep learning models (Resnet18)	The performance of the deep learning model, based on only WSIs, is better than the model based on the primary diagnosis and some demographic variables, such as race and gender, but not as good as age at diagnosis.
Kao	2019	United States	Glioma	Patients treated independently	Overall survival	Radiomic, clinical data	TCIA	347	Deep neural networks, hard negative mining, patch-based 3D U-nets, DeepMedic, SVM classifier with linear kernal	The use of normalized brain parcellation data and tractography data achieved a survival prediction accuracy of ~0.7 on the training data set.
Li	2021	China	Glioma (LGG)	Patients treated independently	Immunotherapy response risk score	Radiomics, immune molecular biomarkers, genetic, clinical data	TCGA	665	Neural network deep learning	Patients at lower risk were more likely to be predicted in the low IMriskScore risk group by the imagingomics deep learning model and have higher survival rates
Mi	2022	United Kingdom	Glioblastoma	Patients treated independently	Overall survival	Radiomic, clinical data	45 from in house glioblastoma data set, 51 from TCGA-GBM data set	132	2D U-net CNN	U-net trained with DL had highest performance and was better than BCEL and HDL for temporalis segmentation to determine cross sectional measurements.
Sun	2021	China	Glioma (LGG)	Patients treated independently	Overall survival	Radiomic, genomic data	TCIA, TCGA	44		Combining MRI and gene expression data in DNN led to more accurate disease specific survival statistics for LGG patients than when tested separately.
Tang	2019	United States	Glioblastoma	Patients treated independently	Overall survival and tumor genotype prediction	Radiomic, genomic biomarker, clinical data	Department of Radiology at University of North Carolina at Chapel Hill	120	Integrated multitask CNN	The combination of imaging phenotype and genotype data input to CNN for OS time prediction for GBM outperformed the mono-task CNN-based and radiomics-based random forest methods.
Wijethilake	2020	Singapore	Glioblastoma	Patients treated independently	Overall survival	Radiomics, genomics	TCGA	59	Hypercolumn-based convolutional network, ANN	Hypercolumn-based CNN Radiogenomic data achieved higher survival prediction accuracies than just radiomic or genomic data alone when predicted using ANN, SVM and linear regression models.

CNN: convolutional neural network; RNN: recurrent neural network; FCNN: fully convolutional neural network; SNN: spiking neural network; ANN: artificial neural network; ML: machine learning; OS; overall survival; LGG: low grade glioma; HGG: high grade glioma; GBM; glioblastoma; SVM: support vector machine; AUC: area under the receiver operating characteristic curve; TCGA: The Cancer Genome Archive; TCIA: The Cancer Image Archive; MRI: magnetic resonance imaging.

**Table 2 cancers-15-00545-t002:** Risk of Bias.

		ROB				Applicability			Overall	
Author	Year	Participants	Predictors	Outcomes	Analysis	Participants	Predictors	Outcomes	ROB	Applicability
Alex	2018	?	+	+	-	+	+	+	-	-
Asthana	2022	?	?	?	?	?	?	?	?	?
Braman	2021	+	+	+	?	+	+	+	?	?
Chaddad	2022	+	+	+	-	+	+	+	-	-
Choi	2021	+	+	+	+	+	+	+	+	+
Fathi	2022	+	+	+	+	+	+	+	+	+
Han	2020	?	+	+	-	+	+	+	-	-
Islam	2021	?	+	+	-	+	+	+	-	-
Jeong	2019	+	+	+	-	+	+	+	-	-
Jiang	2021	+	+	+	+	+	+	+	+	+
Kao	2019	+	+	+	+	+	+	+	+	+
Li	2021	+	+	+	+	+	+	+	+	+
Mi	2022	+	+	+	+	+	+	+	+	+
Sun *	2021	?	?	?	-	?	?	?	-	-
Tang	2019	?	+	+	?	+	+	+	?	?
Wijethilake	2020	?	+	+	+	+	+	+	+	+

ROB = risk of bias; * + indicates low ROB/low concern regarding applicability; - indicates high ROB/high concern regarding applicability; and ? indicates unclear ROB/unclear concern regarding applicability.

## Data Availability

No new data were created in this study. Data sharing is not applicable to this article.

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
