# Peer review of "Multimodal Deep Learning-Based Prognostication in Glioma Patients: A Systematic Review"

_cancers, 2023, doi:10.3390/cancers15020545_

Round 1

Reviewer 1 Report

Revision of systematic review entitled “Multimodal deep learning-based prognostication in brain tumor patients: a systematic review” by Kaitlyn Alleman et al.

The authors aimed to perform a systematic review, using several databases and in accordance with PRISMA guidelines, about the Deep Learning capacity to integrate clinical, biological, and radiological data of brain tumor to improve prognosis prediction. They concluded that studies which compared multimodal models to unimodal-only models had improved predictive performance. Moreover, the use of multimodal data in DL assessments of brain tumors leads to a more accurate overall survival prediction. However, the authors highlighted that the data limitations and a lack of transparency in model and code reporting negatively affected the full extent of multimodal DL as a resource for brain tumor.

The review, based on a relatively low number of scientific report relevant to this topic, is interesting, well conducted and well written. There are some minor criticisms:

1. The authors highlighted how analysis of multimodal models with DL is better than unimodal-only for the prognostic predictivity in human glial tumors. However, the authors should focus more on some essential points in the discussion: 1. There are different types of DL and if they have shown an ability to improve the understanding of complex (multimodal) systems in a single study, it is often difficult to compare the different DL systems and therefore have some comparable or reproducible results. 2. The scientific reports on this topic are still few and unfortunately often include cohorts of patients not fully characterized on the base of clinical and biological features, sometimes very different, also in relation to the type of tumor (high-grade glioma versus low -grade glioma). In this sense, the prognosis, for example, of tumors that can be completely removed compared to those that cannot because in functional areas, completely changes.

2. The authors should also hypothesize possible improvement interventions in this sense: for example, homogeneous databases by histological type, radio-chemo-surgical treatment, molecular characteristics; use of different DLs with different analytical methodologies to test their interchangeability.

Author Response

The authors aimed to perform a systematic review, using several databases and in accordance with PRISMA guidelines, about the Deep Learning capacity to integrate clinical, biological, and radiological data of brain tumor to improve prognosis prediction. They concluded that studies which compared multimodal models to unimodal-only models had improved predictive performance. Moreover, the use of multimodal data in DL assessments of brain tumors leads to a more accurate overall survival prediction. However, the authors highlighted that the data limitations and a lack of transparency in model and code reporting negatively affected the full extent of multimodal DL as a resource for brain tumor.

The review, based on a relatively low number of scientific report relevant to this topic, is interesting, well conducted and well written. There are some minor criticisms:

  1. The authors highlighted how analysis of multimodal models with DL is better than unimodal-only for the prognostic predictivity in human glial tumors. However, the authors should focus more on some essential points in the discussion: 1. There are different types of DL and if they have shown an ability to improve the understanding of complex (multimodal) systems in a single study, it is often difficult to compare the different DL systems and therefore have some comparable or reproducible results. 2. The scientific reports on this topic are still few and unfortunately often include cohorts of patients not fully characterized on the base of clinical and biological features, sometimes very different, also in relation to the type of tumor (high-grade glioma versus low -grade glioma). In this sense, the prognosis, for example, of tumors that can be completely removed compared to those that cannot because in functional areas, completely changes.

Regarding 1: This point has been expanded upon in the Discussion, section 4.1, last paragraph.

Regarding 2: This is an important limitation in the current literature. The Discussion, sections 4.3, final two paragraphs highlight the utility of more comprehensive datasets and advocate for continued efforts in this area.

  1. The authors should also hypothesize possible improvement interventions in this sense: for example, homogeneous databases by histological type, radio-chemo-surgical treatment, molecular characteristics; use of different DLs with different analytical methodologies to test their interchangeability.

This has been addressed in the revised manuscript.

Reviewer 2 Report

I reviewed the manuscript entitled "Multimodal deep learning-based prognostication in brain tumor patients: a systematic review " submitted by Kaitlyn Alleman et al.. This sort of systematic review is useful for readers, particularly those who are not very familiar with these techniques, to grasp the current status of the DL-based prognostic prediction and its limitations. However, I have found several concerns in the manuscript, which hampered the enthusiasm for publishing the manuscript in the journal. Please see below for the details. 

Major points: 

1. The flow of the sentence is overall not well structured. I assume the authors might have tried to be comprehensive to cover various things. As a result, however, the point of the sentence/paragraph became ambiguous. There found a lot of irrelevant descriptions as well. 

2. Study subject: While the authors might have originally intended to study more different types of brain tumors beyond glial tumors, the search result turned out to be only glioma. It will be more helpful for readers to rephrase "brain tumor" with "glioma" in the title and elsewhere. 

Minor points: 

Page 2 Line 38: Regarding "Early detection and diagnosis remains critical and recent advances in tumor characterization and treatment have greatly improved outcomes. ", what disease do the authors intend? If it is about cancers in general, that should be clarified. If it is about malignant glioma, it is not consistent with the subsequent sentence. 

Page 2 Line 39: Regarding the statement "However, prognosis for many tumors remains poor due to a lack of available predictive tools.", it seems to be overstating and incorrect. A more common understanding is that the prognosis of glioma or glioblastoma remains dismal mainly due to a lack of development of modalities for effective treatment and also diagnosis. The lack of available prediction tools should not be the main cause. This sentence should be removed or corrected appropriately. 

Page 3 In Figure 1 (PRISMA Flowchart): It seems to be incomplete regarding the block of "Reports excluded:". Careful double-checking is recommended. 

Page 3 In Figure 1: Regarding the terminologies, are "Reports", "Records", and "Studies" all interchangeable? It is unclear. 

Page 7 Line 150: In the clinical setting, imaging test usually means such as MRI, CT, PET, etc. On the other hand, histopathology slide images are typically treated separately. Therefore, to avoid unnecessary confusion, I would recommend separating them, such as "diagnostic images" and "histologic images." 

Page 8 Line 217: The description of the "Risk of bias" is too few, and not very helpful. 

Page 9 Line 242: "but also for informing therapeutic options and course" - This is overstating. I did not find any rationale to make this claim in the manuscript. This needs to be corrected or omitted. 

Page 9 Line 263: The contents in the section named "4.2. Use in the Clinical Workflow" does not seem to be directly relevant to what the authors reviewed in this study.

Author Response

I reviewed the manuscript entitled "Multimodal deep learning-based prognostication in brain tumor patients: a systematic review " submitted by Kaitlyn Alleman et al.. This sort of systematic review is useful for readers, particularly those who are not very familiar with these techniques, to grasp the current status of the DL-based prognostic prediction and its limitations. However, I have found several concerns in the manuscript, which hampered the enthusiasm for publishing the manuscript in the journal. Please see below for the details. 

Major points: 

  1. The flow of the sentence is overall not well structured. I assume the authors might have tried to be comprehensive to cover various things. As a result, however, the point of the sentence/paragraph became ambiguous. There found a lot of irrelevant descriptions as well. 

This has been addressed in the revised manuscript.

  1. Study subject: While the authors might have originally intended to study more different types of brain tumors beyond glial tumors, the search result turned out to be only glioma. It will be more helpful for readers to rephrase "brain tumor" with "glioma" in the title and elsewhere. 

This has been amended in the revised manuscript.

Minor points: 

Page 2 Line 38: Regarding "Early detection and diagnosis remains critical and recent advances in tumor characterization and treatment have greatly improved outcomes. ", what disease do the authors intend? If it is about cancers in general, that should be clarified. If it is about malignant glioma, it is not consistent with the subsequent sentence. 

This statement has been removed from the revised manuscript.

Page 2 Line 39: Regarding the statement "However, prognosis for many tumors remains poor due to a lack of available predictive tools.", it seems to be overstating and incorrect. A more common understanding is that the prognosis of glioma or glioblastoma remains dismal mainly due to a lack of development of modalities for effective treatment and also diagnosis. The lack of available prediction tools should not be the main cause. This sentence should be removed or corrected appropriately. 

This statement has been amended in the revised manuscript.

Page 3 In Figure 1 (PRISMA Flowchart): It seems to be incomplete regarding the block of "Reports excluded:". Careful double-checking is recommended. 

This has been amended in the revised manuscript.

Page 3 In Figure 1: Regarding the terminologies, are "Reports", "Records", and "Studies" all interchangeable? It is unclear. 

This has been amended in the revised manuscript.

Page 7 Line 150: In the clinical setting, imaging test usually means such as MRI, CT, PET, etc. On the other hand, histopathology slide images are typically treated separately. Therefore, to avoid unnecessary confusion, I would recommend separating them, such as "diagnostic images" and "histologic images." 

This has been amended the revised manuscript.

Page 8 Line 217: The description of the "Risk of bias" is too few, and not very helpful. 

As noted in line 91, PROBAST uses 20 signaling questions across 4 key domains to determine risk of bias of studies which develop clinical outcome prediction models.  

Page 9 Line 242: "but also for informing therapeutic options and course" - This is overstating. I did not find any rationale to make this claim in the manuscript. This needs to be corrected or omitted. 

This statement was omitted in the revised manuscript.

Page 9 Line 263: The contents in the section named "4.2. Use in the Clinical Workflow" does not seem to be directly relevant to what the authors reviewed in this study.

We feel that “Use in the Clinical Workflow” and the following section “Challenges for Implementation” discusses ways that DL modeling is already being used in other areas of oncology and provides added value to the reader.

Reviewer 3 Report

The abstract should mention significance of your study, like why this topic is important, method used why etc.

In the introduction, what key theoretical perspectives and empirical findings in the main literature have already informed the problem formulation? What major, unaddressed puzzle, controversy, or paradox does this research address? 

As this is a review paper, you must expand your introduction and literature review section. 

More explanation needed on how authors selected the papers?

Below papers have some interesting implications that you could discuss in your introduction and how it relates to your work.

Vulli, A.; et al.. Fine-Tuned DenseNet-169 for Breast Cancer Metastasis Prediction Using FastAI and 1-Cycle Policy. Sensors 2022, 22, 2988.

El-Sappagh, Shaker, et al. "Automatic detection of Alzheimer’s disease progression: An efficient information fusion approach with heterogeneous ensemble classifiers." Neurocomputing (2022

Why authors focused only on Multimodal deep learning-based prognostication?

What are the limitations of the present work?

What are some key contributions of the present study?

What are the practical implications of this research

Author Response

The abstract should mention significance of your study, like why this topic is important, method used why etc.

This is stated in the first 3 lines of the abstract.

In the introduction, what key theoretical perspectives and empirical findings in the main literature have already informed the problem formulation? What major, unaddressed puzzle, controversy, or paradox does this research address? 

This is discussed in the last paragraph of the introduction.

As this is a review paper, you must expand your introduction and literature review section. 

Our manuscript represents a systematic review of the literature, within the given inclusion criteria to make informed conclusions.  As such, all available literature is discussed at present.

More explanation needed on how authors selected the papers?

Inclusion criteria are discussed in the methods section and outlined in Figure 1.

Below papers have some interesting implications that you could discuss in your introduction and how it relates to your work.

Vulli, A.; et al.. Fine-Tuned DenseNet-169 for Breast Cancer Metastasis Prediction Using FastAI and 1-Cycle Policy. Sensors 2022, 22, 2988.

El-Sappagh, Shaker, et al. "Automatic detection of Alzheimer’s disease progression: An efficient information fusion approach with heterogeneous ensemble classifiers." Neurocomputing (2022

These suggestions have been incorporated into the introduction of the revised manuscript.

Why authors focused only on Multimodal deep learning-based prognostication?

As detailed in the introduction of the manuscript, multimodal DL is superior to unimodal analysis and represents a more comprehensive and useful approach for future applications.

What are the limitations of the present work?

This is discussed in the Limitations (4.4) section of the manuscript.

What are some key contributions of the present study?

This is discussed in the Conclusions section (5) of the manuscript.

What are the practical implications of this research?

This is discussed in the Use in the Clinical Workflow (4.2) and Challenges for Implementation in Clinical Workflow (4.3) sections of the manuscript.

Round 2

Reviewer 3 Report

.

Author Response

There are no additional comments to address below.